# Pitaya Juice Consumption Protects against Oxidative Damage Induced by Aflatoxin B1

**DOI:** 10.3390/jof9090874

**Published:** 2023-08-24

**Authors:** Luiggi Müller Madalosso, Franciéle Romero Machado Balok, Vandreza Cardoso Bortolotto, Mustafa Munir Mustafa Dahleh, Lucas Gabriel Backes, Elizabeth Sabryna Sarquis Escalante, Fernanda Vilhalba Benites, Francisco Andrey da Silva e Silva, Hecson Jesser Segat, Silvana Peterini Boeira

**Affiliations:** 1Laboratory of Pharmacological and Toxicological Evaluations Applied to Bioactives Molecules—LaftamBio, Federal University of Pampa, Itaqui 97650-000, Brazil; luiggimadalosso.aluno@unipampa.edu.br (L.M.M.); lucasbackes.aluno@unipampa.edu.br (L.G.B.); elizabethescalante.aluno@unipampa.edu.br (E.S.S.E.); fernandabenites.aluno@unipampa.edu.br (F.V.B.); franciscosilva.aluno@unipampa.edu.br (F.A.d.S.e.S.); hecsonsegat@unipampa.edu.br (H.J.S.); 2Programa de Pós-Graduação em Bioquímica, Federal University of Pampa, Uruguaiana 97650-000, Brazil; francielemachado.aluno@unipampa.edu.br (F.R.M.B.); vandreza.bortolotto@gmail.com (V.C.B.); mustafadahleh.aluno@unipampa.edu.br (M.M.M.D.)

**Keywords:** hepatotoxicity, antioxidant effects, mycotoxin

## Abstract

Mycotoxins are toxic fungal metabolites and are responsible for contaminating several foods. The intake of foods contaminated by these substances is related to hepatotoxicity and carcinogenic effects, possibly due to increasing oxidative stress. The current study evaluated Pitaya fruit juice’s antioxidant effects on oxidative damage aflatoxin B1 (AFB1)-induced. Rats received 1.5 mL of Pitaya juice via gavage (for 30 days), and on the 31st day, they received AFB1 (250 µg/kg, via gavage). Forty-eight hours after the AFB1 dose, rats were euthanized for dosages of alanine transaminase (ALT), aspartate aminotransferase (AST), and alkaline phosphatase (ALP); dosage of oxidative markers (thiobarbituric acid reactive species (TBARS), reactive species (RS)) and antioxidant defenses (catalase (CAT), superoxide dismutase (SOD), Glutathione S-transferase (GST) activities and Glutathione (GSH)) levels in the liver; and detection of Heat shock protein 70 (Hsp-70) and nuclear factor- erythroid 2-related factor 2 (Nrf2) immunocontent in the liver. Our results indicated that the Pitaya juice reduced ALP activity. Further, rats exposed to AFB1 experienced liver damage due to the increase in TBARS, RS, and Hsp-70 and the reduction in CAT, GSH, and Nrf2. Pitaya juice could, however, protect against these damages. Finally, these results indicated that pre-treatment with Pitaya juice was effective against the oxidative damage induced. However, other aspects may be elucidated in the future to discover more targets of its action against mycotoxicosis.

## 1. Introduction

Mycotoxins are toxic substances produced by fungal secondary metabolism, consisting of the most severe food safety problems due to their effects on human and animal health [1,2]. These secondary metabolites are derived from central metabolic pathways and primary metabolite pools, with acyl-CoAs as building blocks in synthesizing polyketides and aflatoxins [3].

Contamination and exposure to mycotoxins is of concern. Recent review has shown that as much as 60% to 80% of global food production may be contaminated by mycotoxins [4]. Among mycotoxins, aflatoxins can contaminate many products, including peanuts and derivatives, spices, tree nuts, corn, crude vegetable oil, cottonseed, fermented beverages made from grains, milk and derivatives, cocoa beans, nuts, and cereals [5].

When metabolized by the body, the aflatoxins generate hepatotoxic metabolites which present high toxicity and thermostability, act at low concentrations, and induce carcinogenic, mutagenic, and teratogenic effects [6,7,8,9,10]. The aflatoxin B1 (AFB1) is metabolized into a reactive metabolite (AFB1-8-9-epoxide) in the liver by cytochrome P450 enzymes; it harmful to the organism because it causes carcinogenic, mutagenic, and teratogenic effects, it reduced protein synthesis, and it causes the inhibition of blood clotting potentials [11,12,13].

Oxidative stress is a common manifestation of AFB1-induced toxicity. [14]. Oxidative stress occurs when the production of free radicals and reactive species exceeds the body’s ability to neutralize them through its defense system [15,16]. Several studies have found a link between the toxic effects of AFB1 and the observed oxidative stress, indicating hepatotoxicity caused by the mycotoxin [14,17]. In this context, understanding the structure, toxicity, and molecular mechanisms are critical to avoid contamination and pursue the complete detoxification of the aflatoxins. Furthermore, several studies using bioactive molecules have been conducted to reduce the harmful effects of mycotoxin on health [18,19].

The Pitaya or “Dragon Fruit” (*Hylocereus* spp.) is a fruit native to Central and South America [20]. These fruits produce antioxidant compounds, such as betalains, responsible for color, and prebiotic oligosaccharides. Further, these fruits contain polyphenols, flavonoids, and vitamin C, which enhance their antioxidant potential [21,22]. 

It was previously reported that Pitaya consumption reduces total cholesterol, triglycerides, and LDL levels while increasing HDL levels [23]. In addition, it is related to improved insulin resistance and antihyperlipidemic, antimicrobial, antidiabetic, and hepatoprotective activities [24].

Based on the literature, AFB1 can cause extensive oxidative damage to the human organism. For this, research demonstrated that the antioxidant function of natural substances such as Pitaya are significant findings against the harmful effects of mycotoxins. In this sense, the main objective of this study was to assess if the antioxidant properties of Pitaya juice could mitigate or avert the oxidative damage and hepatotoxicity caused by AFB1. In short, our findings reveal that Pitaya juice effectively guards against AFB1-induced hepatotoxicity and cellular oxidative damage.

## 2. Materials and Methods

### 2.1. Preparation of Red Pitaya Juice

The Pitaya juice was prepared using red Pitaya (*Hylocereus polyrhizus*) from producers on a farm located in the interior of the municipality of Itaqui-Rio Grande do Sul in Brazil (“Pitayas Tapera do Coqueiro” located in Fundo Grande-RS 529). For the preparation of the juice, fruits harvested at the stage of commercial maturation with a uniform pink color, were used. The fruits were initially cleaned and peeled and then crushed and homogenized. After obtaining the pulps, a concentration (1:1) was prepared, which was diluted with distilled water. Finally, the juice was stored at −20 °C for later use.

### 2.2. Drugs and Reagents

The mycotoxin AFB1 (Cas. No. 1162-65-8; ≥95% purity), 2′,7′-Dichlorodihydrofluorescein diacetate (DCFH-DA), thiobarbituric acid (TBA), epinephrine, glutathione (GSH), 1-chloro-2,4 dinitrobenzene (CDNB), O-phthalaldehyde (OPA), β-Nicotinamide adenine dinucleotide phosphate (NADPH, Tetrasodium Salt), anti-rabbit heat shock protein 70 (Hsp-70), and anti-mouse β-actin e 3′,5,5′-Tetramethylbenzidine (TMB) were obtained from Sigma (St. Louis, MO, USA). The other chemical reagents were obtained from commercial suppliers, which are available at the Laftambio Pampa Laboratory, UNIPAMPA Campus, Itaqui.

### 2.3. Animals

To conduct the experimental design, adult male Wistar rats (90 days old) from the vivarium of the Federal University of Santa Maria (UFSM) were used. The animals were housed in polypropylene cages under controlled conditions of light (12 h light/dark cycle), controlled temperature (25 ± 2 °C), and access to food and water ad libitum. Before the experiments began, the animals were habituated to maintenance conditions for one week to avoid environmental stress. All procedures were conducted in accordance with the guidelines of the Committee on Care and Use of Experimental Animal Resources, and this research was approved by the Ethics Committee on Animal Use (CEUA) of the Federal University of Pampa (Number 023/2022). All necessary measures were taken to minimize the pain of the experimental animals.

### 2.4. Experimental Design and Pre-Treatment

After the acclimatization period, the rats were randomly divided into four groups (n = 6 animals per group): Group Control (1.5 mL of water); Group Pitaya (1.5 mL of Pitaya juice); Group AFB1 (1.5 mL of water + AFB1); and Group Pitaya + AFB1 (1.5 mL of Pitaya juice + AFB1). The experimental protocols were elaborated to keep the minimum number of animals necessary to obtain the results.

Initially, pre-treatment with 1.5 mL of Pitaya juice was conducted for 30 consecutive days in the respective Pitaya and Pitaya + AFB1 groups (intragastric gavage). The Control and AFB1 groups received 1.5 mL of water (vehicle) during this initial period of the experiment. On the 31st day, the AFB1 and Pitaya + AFB1 groups received a single dose of mycotoxin (250 µg/Kg) via intragastric gavage. The animals in the other groups received DMSO 0.02%, used to prepare a stock solution of AFB1 as previously described [25]. After 48 h of administration of AFB1 or DMSO, the rats were euthanized with pentobarbital (180 mg/kg, intraperitoneally) to proceed with the collection of blood and tissues. Blood samples were collected through cardiac puncture, transferred to tubes containing the anticoagulant (heparin) and then centrifuged at 3000 r.p.m. for 10 min to obtain the serum. Blood serum and clean, dissected liver were stored at −80 °C for further analysis, including liver damage markers, biochemical analysis of oxidative stress and Western blotting. The experimental design is shown below in Figure 1.

The liver samples were quickly collected and homogenized in Tris-HCl 50 mM (pH 7.4) buffer in a 1:10 (*w*:*v*) proportion and then centrifuged in 2500 r.p.m. for 10 min. Supernatants were used for measurements of the oxidative stress analyses, except for the Glutathione (GSH), homogenized in perchloric acid (HClO4) 0.1 M in a 1:10 (*w*:*v*) proportion and then centrifuged at 3000 r.p.m. for 10 min to obtain the supernatant.

### 2.5. Liver Damage Markers

Blood serum was used for biochemical determinations of alanine aminotransferase (ALT), aspartate aminotransferase (AST), and alkaline phosphatase (ALP) activities using kits according to the manufacturer’s instructions (Labtest, Diagnostica S.A., Minas Gerais, Brazil). The enzymatic activities were expressed as U/L.

### 2.6. Parameters of Oxidative Stress

#### 2.6.1. Thiobarbituric acid Reactive Species (TBARS)

Lipid peroxidation levels were measured in the liver using thiobarbituric acid reactive species (TBARS). According to the method described by Ohkawa et al. [26], malondialdehyde (MDA), which is a final product of the peroxidation of fatty acids, interacts with thiobarbituric acid for the color reaction to occur. The reaction involves reagents, such as liver supernatant, acetic acid buffer (2.5 M, pH 3.4), thiobarbituric acid (TBA) 0.8% (pH 3.2), SDS 8.1%, and distilled water. Finally, the reaction mixture was incubated at 95 °C for 2 h in a water bath, using a glass ball to condense. After 2 h and cooling, the absorbance at 532 nm was measured. Results are expressed in MDA (nmol/g tissue).

#### 2.6.2. Reactive Species Levels (RS)

RS levels were quantified using liver supernatant (S1), following the protocol by Pérez-Severiano et al. [27]. In the analysis, DCFH-DA is a chemically reduced form of fluorescein (RS Indicator), which is converted into 2′,7′ dichlorofluorescein (DCF), emitting fluorescence after oxidation.

The liver supernatant, 10 mM Tris HCl buffer pH 7.4, and DCFH-DA (1 mM) were incubated for 1 h. Then, the reading was performed with an excitation length of 488 nm and emission of 530 nm in the Cary Eclipse Fluorescence spectrophotometer (Agilent Technologies). The fluorescence intensity is related to RS production, in which the results were expressed as a percentage in relation to the control group.

#### 2.6.3. Enzyme Activity of Catalase (CAT)

The enzymatic activity of the CAT, responsible for neutralizing the substrate hydrogen peroxide (H_2_O_2_), was measured. According to the methodology proposed by Aebi [28], it measured the disappearance of H_2_O_2_ at 240 nm for 120 s in a thermostated spectrophotometer (37 °C). For the reaction, it used the liver supernatant (S1, diluted 1:10) and H_2_O_2_ substrate in a medium with potassium phosphate buffer (50 mM pH 7.0). Expression of enzyme activity was demonstrated in Units (U)/mg of protein.

#### 2.6.4. Enzyme Activity of Superoxide Dismutase (SOD)

According to the method previously described by Misra and Fridovich [29], the activity of the superoxide dismutase (SOD) was spectrophotometrically measured in the hepatic tissue. The protocol evaluates the ability of the enzyme to inhibit the autoxidation of epinephrine to adrenochrome. The supernatant was inserted into the medium containing the sodium carbonate buffer (Na_2_CO_3_; 57.7 mM) and the kinetic analysis started after the addition of epinephrine (6 mM), with the color reaction measured at 480 nm.

Each enzyme unit was described as the amount of enzyme to inhibit the rate of epinephrine autooxidation by 50% at a temperature of 30 °C. The results for SOD enzymatic activity were expressed in U/mg of protein.

#### 2.6.5. Enzymatic Activity of Glutathione-S-Transferase (GST)

The reaction presents an aliquot of liver supernatant (diluted 1:20), 0.1 M potassium phosphate buffer (pH 7.5), 100 mM GSH (1 mM), and CDNB (1 mM), which was used as substrate. Enzyme activity in liver tissue was measured at 340 nm for 2 min, using the method previously described by Habig et al. [30]. The method is based on the conjugation of glutathione with CDNB (substrate), with results showed as CDNB (nmol/min/mg of protein).

#### 2.6.6. Fluorimetric Assay of Reduced Glutathione (GSH)

We proceeded with the quantification of GSH levels according to the methodology of Hissin and Hilf [31], with adaptations of Sies [32] and Forman et al. [33]. Exclusively for this analysis, the liver was weighed and homogenized in perchloric acid (HClO_4_ 0.1 M) in the proportion (1/10). From the supernatant fraction (S1), 100 µL of tissue supernatant was added to 800 µL 0.1 M potassium phosphate buffer (TFK) pH 8 and finally with 100 µL of OPA as fluorophore (1 mg/mL).

Then, the mixture was incubated for 15 min at room temperature, reading at a wavelength of 420 nm (emission) and 350 nm (excitation) on a Cary Eclipse Fluorescence spectrophotometer (Agilent Technologies). At the same time, a calibration curve was made for the calculations, in which these results were expressed as GSH (nmol/g tissue).

### 2.7. Western Blot Analysis

Western blot analysis was conducted with slight modifications following the protocol previously described by Guerra et al. [34]. Briefly, rats were euthanized, and the liver was homogenized in an ice-cold buffer containing KCl (10 mM), MgCl_2_ (2 mM), EDTA (1 mM), NaF (1 mM), aprotinin (10 µg/mL), β-glycerophosphate (10 mM), PMSF (1 mM), DTT (1 mM), and sodium orthovanadate (2 mM) in HEPES (10 mM, pH 7.9). The samples were incubated on ice for 15 min and then centrifuged at 16,000× *g* for 45 min at 4 °C. The supernatant was collected for further processing, and the protein concentration was determined using the Bradford method [35]. Equivalent amounts of protein (80 µg) were mixed with a concentrated loading buffer consisting of Tris (200 mM), glycerol (10%), SDS (2%), β-mercaptoethanol (2.75 mM), and bromophenol blue (0.04%) in a 0.2:1 ratio and boiled for 10 min.

The proteins were separated using 12% SDS–PAGE, and the gels were transferred onto Amersham™ Protran^®^ Premium Western blotting nitrocellulose membranes using the Transfer-Blot^®^ Turbo™ Transfer System (1.0 mA; 30 min). Protein transfer was confirmed by staining the membranes with ponceau 0.5% [36]. After blocking the membranes with 1% BSA (Bovine Serum Albumin) in TBS-T (0.05% Tween 20 in Tris-borate saline), the blots were incubated overnight with specific primary polyclonal antibodies, including anti-mouse β-actin (loading control) (1:1000; Sigma-Aldrich^®^; St. Louis, MO, USA), anti-mouse nuclear factor erythroid 2–related factor 2 (Nrf2) (1:1000; Santa Cruz Biotechnology, Inc., Santa Cruz, CA, USA), and anti-rabbit Hsp-70 (1:1000; Sigma-Aldrich^®^; St. Louis, MO, USA). The membranes were then washed three times with TBS-T and incubated with horseradish peroxidase-conjugated secondary antibodies (1:5000, anti-mouse IgG-HRP or anti-rabbit IgG; Santa Cruz Biotechnology, Inc., Santa Cruz, CA, USA) for 2 h. After the incubation period with the secondary antibody, the membranes were subjected to a triple wash with Tris-Buffered Saline with Tween^®^ 20 (TBST) and then with the TMB reagent (Sigma-Aldrich^®^; St. Louis, MO, USA). Protein bands were visualized using TMB through color development, making it possible to observe the antibody reaction.

Finally, the membranes were dried, scanned, and quantified. For the quantification process, the ImageJ PC version software (NIH, Bethesda, MD, USA) was used, highlighting that the quantification covered the bands corresponding to the study antibodies (Nrf2 and Hsp-70), together with the β-actin protein as control. Values derived from software quantification were incorporated into an antibody/β-actin ratio for each sample. This approach enables a standardized protein control mechanism, mitigating discrepancies resulting from individual animal variations. The results were normalized by arbitrarily setting the ratio between the protein band and β-actin loading control.

### 2.8. Statistical Analysis

The data were subjected to a normality test (Shapiro–Wilk test). All data submitted to a *two-way* analysis of variance (ANOVA), followed by Tukey’s post hoc test. We used AFB1 and Pitaya juice as independent variables. Data were expressed as the mean and standard error of the mean (S.E.M.). Statistical analysis and figures were made using GraphPad Prism 9.0 software. *p* values less than 0.05 (*p* < 0.05) were considered significant.

## 3. Results

### 3.1. Pitaya Juice Partially Attenuates AFB1-Induced Liver Toxicity

AST activity is demonstrated in Figure 2A (F_(1, 14)_ = 2.73; *p* = 0.12), where exposure to AFB1 significantly elevated plasma AST enzyme activity compared to control. Furthermore, a significant increase is also shown in ALT activity in Figure 2B (F_(1, 12)_ = 2.29; *p* = 0.15). For both liver enzymes, AST and ALT (Figure 2A,B; *p* < 0.05), their increase in plasma was observed, indicating liver toxicity after exposure to AFB1.

ALP levels were significantly elevated in the AFB1 group compared to the Pitaya+ AFB1 group (*p* < 0.01). In Figure 2C (F_(1, 17)_ = 4.58; *p* = 0.04), ingestion of the Pitaya juice prevented the increase in this marker due to liver damage, protecting the group from AFB1 toxicity. Indeed, two-way ANOVA revealed an interaction only in ALP activity between Pitaya + AFB1 administration.

### 3.2. Pitaya Pre-Treatment Protected Rats from Lipid Peroxidation and AFB1-Induced RS

In Figure 3A (F_(1, 12)_ = 4.65; *p* = 0.05), the exposure to a single dose of AFB1 caused an increase in MDA levels compared to control (*p* < 0.05). The animals that received pre-treatment with Pitaya juice (Pitaya + AFB1) showed reduced levels of MDA (*p* < 0.05). This result confirmed that Pitaya exerted a protective effect against lipid peroxidation, significantly attenuating the effects caused by the AFB1.

As for RS levels, the AFB1 group showed a significant increase in this parameter compared to the control (*p* < 0.05), as shown in Figure 3B (F_(1, 16)_ = 5.74; *p* = 0.02). Again, pre-treatment with Pitaya juice was effective against the oxidative stress caused after exposure to AFB1, as it significantly reduced the RS generated, preventing the increase in RS in the Pitaya + AFB1 group (*p* < 0.05).

### 3.3. Pitaya Attenuated the CAT Enzymatic Activity Altered by AFB1 but Not Significantly

Compared to control group, CAT enzyme activity was significantly reduced in the AFB1 group (*p* < 0.01). While in the other groups, there was no statistical significance, as shown in Figure 4A (F_(1, 12)_ = 7.60; *p* = 0.01). On the other hand, the data presented in the SOD analysis do not show a significant difference in SOD activity in all comparisons made, as shown in Figure 4B (F_(1, 16)_ = 2.40; *p* = 0.14).

### 3.4. AFB1 Alters GST and GSH, with Protective Effect of Pitaya on GST

In the enzymatic activity of the GST that is responsible for the detoxification of xenobiotics, the group exposed to AFB1 showed a significant increase compared to the control (*p* < 0.01), as shown in Figure 5A (F_(1, 16)_ = 20.23; *p* < 0.0004). On the other hand, the rats that consumed Pitaya juice (Pitaya +AFB1) showed lower activity of the GST enzyme compared to the AFB1 group (*p* < 0.01). The intake of Pitaya juice prevented the increase in GST activity that occurred only with AFB1.

According to Figure 5B (F_(1, 16)_ = 1.89; P = 0.18), the AFB1 and Pitaya + AFB1 groups showed reduced levels of GSH in the liver when compared to the Control (*p* < 0.001) and Pitaya (*p* < 0.05) groups, respectively. However, the amount and period of pre-treatment with Pitaya were not able to fully reverse the changes in GSH.

### 3.5. Pitaya Modulated the Expression of Hsp-70 and Nrf2 Markers in Mice Exposed to AFB1

Heat Shock Proteins (HSPs) are induced in response to physiological and environmental insults, while Nrf2 is a transcription factor that regulates cellular defense against toxic and oxidative insults. They constitute two proteins with roles in pathways related to oxidative stress. The expression levels of the Hsp-70 and Nrf2 markers in the hepatic tissue were quantified using the Western Blotting technique.

It was observed that the animals exposed to a single dose of 250 µg/Kg of AFB1 showed a significant increase in Hsp-70 protein expression compared to the control group (P< 0.0001), as observed in Figure 6A (F_(1, 20)_ = 11.86; *p* = 0.0026). Exposure to AFB1 probably provoked insults to the hepatic tissue, causing an increase in Hsp-70. The animals that received pre-treatment for 30 days with Pitaya juice (Pitaya + AFB1 group) showed lower Hsp-70 protein expression even when exposed to AFB1 (*p* < 0.001).

In the expression of Nrf2, it was observed that the animals exposed to AFB1 had lower protein expression compared to the control group, as shown in Figure 6B (*p* < 0.001). Again, ingestion of Pitaya juice before exposure to AFB1 protected against damage involving exposure to mycotoxin AFB1 in Nrf2 immunocontent (F_(1, 20)_ = 4.84; *p* = 0.03), since the Pitaya+ AFB1 group showed greater expression (*p* < 0.01). Thus, pre-treatment for 30 days with Pitaya juice showed a protective effect since it reversed both Hsp-70 and Nrf2 changes in the liver tissue of rats exposed to AFB1.

## 4. Discussion

In the current study, it was demonstrated that AFB1 administration induced several deleterious changes in animals. Pre-treatment with Pitaya juice revealed a potential antioxidant effect, by attenuating markers of oxidative stress and protecting against AFB1-mediated hepatotoxicity. We emphasize that AFB1 is a highly toxic mycotoxin that contaminates food products [37]. Thus, to understand the antioxidant role of Pitaya in living organisms after exposure to AFB1, we investigated the plasmatic activity of enzymes related to hepatotoxicity. Further, we evaluated the levels of RS, lipid peroxidation, antioxidant enzymes, and expression of proteins related to oxidative stress in the liver of rats. This set of results was necessary to elucidate the protective effects of Pitaya juice.

The exposure to AFB1 resulted in a significant increase in plasmatic AST, ALT, and ALP activities. The plasmatic increase in these enzymes is related to hepatobiliary damage; while ALP indicates a dysfunction in the bile ducts, AST and ALT activities are increased when there is damage to hepatocytes [38,39]. This event occurs because noxious stimuli result in cell membrane rupture and the release of these enzymes into the blood [39]. In this sense, our results indicate the effectiveness of the AFB1 mycotoxin exposure protocol in causing hepatobiliary changes. Further, it is already described in the literature that prolonged exposure to AFB1 causes hepatocyte pyroptosis and oxidative stress, resulting in liver damage [40]. In addition, in this study, it was found that pre-treatment with Pitaya juice induced a significant reduction in ALP but not in ALT and AST activities. Corroborating with our outcomes, a study conducted by Mahmood et al. [41] demonstrated that pre-treatment with an isolated flavonoid compound reduced ALP activity and, further, the liver damage induced by paracetamol. In contrast, in another study, the AFB1-exposed group showed higher ALP levels than the control group, indicating biliary pathway compromise [42]. We probably only found the protective effect of Pitaya juice in ALP activity because the period of exposure and the doses of Pitaya juice used in the experimental protocol were not enough to compensate for the severity of the damage caused to the liver enzymes.

We also quantified the levels of RS and MDA in the liver tissue of rats. We performed these assays because AFB1 causes an increase in RS [43] and oxidative damage to hepatic tissue [44]. These damages occur when there is an imbalance between antioxidant and oxidant agents, resulting in oxidative stress. The RS is highly reactive interacting with cell membrane electrons, causing cell damage and lipid peroxidation, with potential cytotoxicity and cell death [45]. Our results indicate that AFB1 exposure increased the RS and MDA levels. As described, RS generation can cause cell damage, as proven by and increase in MDA, as MDA is a compound generated when there is lipid peroxidation [46,47]. On the contrary, we found that Pitaya juice pre-treatment prevented an increase in the RS and MDA levels, reinforcing the antioxidant effect of the treatment against AFB1-induced liver damage. Corroborating with our results, a study conducted by Zhang et al. [48] showed morphological lesions and oxidative damage in chicken livers exposed to AFB1. In this same study, the groups which received a diet enriched with curcumin (another antioxidant compound) showed a reduction in AFB1-induced harmful effects.

Regarding oxidative stress, to eliminate RS and minimize damage to tissues, the organism counts with an antioxidant system divided into enzymatic and non-enzymatic defenses. CAT, SOD, and GST are enzymes antioxidants; and the vitamins E, C and GSH are non-enzymatic ones [49]. CAT catalyzes the oxidation of hydrogen (H_2_O_2_) in water and oxygen, preventing the formation of free radicals. The SOD catalyzes the conversion of superoxide (O_2_^•−^) to hydrogen peroxide (H_2_O_2_) and oxygen (O_2_), which is necessary for the removal of RS from cells [50]. The present study showed that AFB1 exposed rats showed a reduced activity of the CAT when compared to the control group, evidencing the harmful effects of this mycotoxin to the liver. Pre-treatment with Pitaya juice does not alter the activity of CAT and neither AFB1 or Pitaya juice altered SOD activity. Another study using AFB1 in chickens also indicated a decrease in CAT activity for the mycotoxin-exposed group [48]. Furthermore, as described in the study by Li et al. [51], who verified the protective effects of Curcumin on exposure to AFB1, there was a significant reduction in the enzymatic activity of CAT in the liver. Therefore, this finding also corroborates our study, as we found less activity after exposure to AFB1. Although the activity of these antioxidant enzymes was not strongly altered, the RS generation and MDA levels were reduced in the Pitaya-pretreated group. These results show that the antioxidant enzyme activity could be preserved, and the antioxidant compounds (vitamins and natural pigments, such as flavonoids, carotenoids, and anthocyanins) of juice could exert the role of scavenger [52,53].

As demonstrated in the results, Pitaya juice attenuated the toxicity to the liver, observed by the normalization of the enzymatic activity of the GST. Exposure to AFB1 considerably elevated GST levels in the liver. According to Wang et al. [40], in their research with curcumin, there was an upregulation of molecules related to the detoxification of xenobiotics, such as GST and GSH, in mice exposed to mycotoxin AFB1. In our study, exposure to AFB1 caused lower levels of GSH, similar to the research by Li et al. [51], where AFB1 caused a reduction in the GSH content, indicating oxidative stress in the liver. However, while compounds such as curcumin restored GSH levels, Pitaya juice at the concentration used was not able to significantly reverse these levels in the liver tissue of rats. GSH is found abundantly in tissues of the liver and kidneys, and it is necessary to maintain adequate levels for homeostasis of the redox state, cell signaling, physiological functions, etc.

Finally, regarding the expression of the levels of Hsp-70 and Nrf2 markers, it was clarified that the 30 day pre-treatment with Pitaya juice was able to minimize the oxidative stress damage caused by AFB1. Heat shock transcription factors (HSFs) and Nrf2 are involved in the oxidative processes. The Hsp-70 expression is induced by oxidative stress as a cytoprotective function to prevent oxidative damage [54]. Our outcomes showed an increase in this protein in the AFB1 group while Pitaya juice prevented its increase, suggesting, together with other results described above, the antioxidant effect of our treatment. Further, corroborating to our result, a study conducted by Harahap et al. [55], demonstrated the function of the extract of the red pulp of the Pitaya (100 to 300 mg/kg) in reducing the expression of Hsp-70 in rats with a strenuous exercise protocol.

Another protein involved in the oxidative stress process is Nrf2, which plays a role in cell survival [56]. Our results indicated that the expression of Nrf2 was significantly reduced with AFB1 exposure, while pre-treatment with Pitaya positively regulated Nrf2 expression. The Nrf2 is a factor responsible for inducing the expression of cytoprotective genes, which can generate enzymes involved in phase I, II, and III of drug detoxification, with the elimination of pro-oxidants to maintain cellular homeostasis [57].

According to Wan et al. [58], in a study with ducklings which received Curcumin treatment for 21 days, there was a reduction in AFB1-induced spleen damage. It was found that curcumin activated the Nrf2 signaling pathway and regulated the expression of antioxidant enzymes. In another study involving AFB1 and the Nrf2 pathway, the protective effects of the curcumin (100 or 200 mg/kg for 30 days) against AFB1-induced liver damage [41]. Just as the Nrf2 pathway is described as a potential therapeutic target, the same can be observed for Pitaya juice, since pre-treatment also regulates Nrf2 expression.

Like other substances, Pitaya upregulated the expression of Nrf2 and associated antioxidant molecules such as CAT and GST. It is worth mentioning that no data were found in the literature representing the effect of Pitaya and AFB1. We represent other natural substances with similar performance to Pitaya. According to Zhou et al. [59] who evaluated the effects of aflatoxin B1 and resveratrol on the viability of the MAC-T cow mammary epithelial cell line, resveratrol alleviated the cytotoxicity induced by aflatoxin B1 (increased ROS and decreased the expression of mRNA transcripts that include the Nrf2, SOD2, and HSP70). After 6 h, SOD2 and Nrf2 expression decreased in all groups, suggesting that resveratrol can antagonize AFB1 without causing a toxic stress response. These results differ in that a reduction in Nrf2 expression was found, which is consistent with AFB1 toxicity.

It was previously demonstrated by Sun et al. [60] that plant flavonoids can play a fundamental role against pathologies, acting through pathways such as Nuclear factor kappa light chain enhancer of activated B cells (NF-κB). It is known that AFB1 induces apoptosis in microglial cells through oxidative stress, activating the NF-κB signaling pathway. The insertion of Pitaya as an antioxidant is important for neutralizing oxidative stress and reducing apoptosis, probably due to its performance in pathways such as NF-κB [61].

## 5. Conclusions

The study’s findings confirm Pitaya juice’s protective effects against AFB1 mycotoxin-induced oxidative damage, leading to oxidative stress and hepatotoxicity. This is demonstrated through various markers such as the enzymatic activity of ALP in plasma, reduced RS and TBARS levels, regulated enzyme activity (CAT and GST), and increased GSH levels in the liver. Additionally, Pitaya juice’s role in modulating protein expression involved in the redox state (Hsp-70 and Nrf2) showcases its antioxidant capabilities across different cellular contexts. As we continue to explore novel pathways and mechanisms of action that contribute to Pitaya’s antioxidant potential, further investigations could reveal additional targets for its action against mycotoxicosis. This aligns with our group’s initial focus on the impact of Pitaya juice on AFB1-induced damage.

## Figures and Tables

**Figure 1 jof-09-00874-f001:**
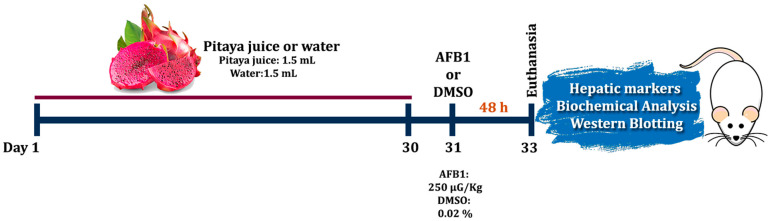
Experimental design: The rats received the Pitaya juice/water for 30 days. Then, the AFB1 in a dose of 250 µg/kg of body weight was administered via gavage. After 48 h of AFB1 administration all animals were euthanized, and the blood and liver samples were collected for further analysis.

**Figure 2 jof-09-00874-f002:**
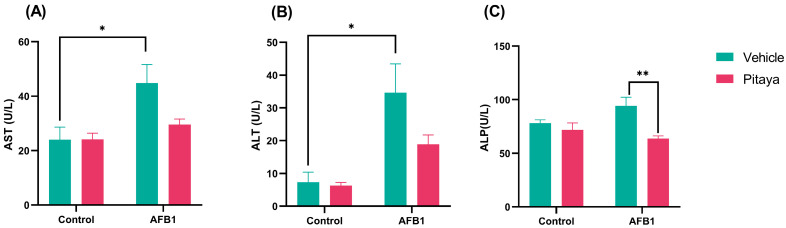
Effect of the Pitaya juice pre-treatment on enzymatic activity of hepatic parameters (**A**) AST, (**B**) ALT, and (**C**) ALP in rats’ plasma exposed to AFB1. The significance was determined through variance analysis (ANOVA, two-way) with post hoc Tukey’s range test. Data are expressed as mean ± S.E.M. (n = 4–5). The values are considered significant when *p* < 0.05. * indicates a significance level for *p* < 0.05 and ** indicate a significance level for *p* < 0.01.

**Figure 3 jof-09-00874-f003:**
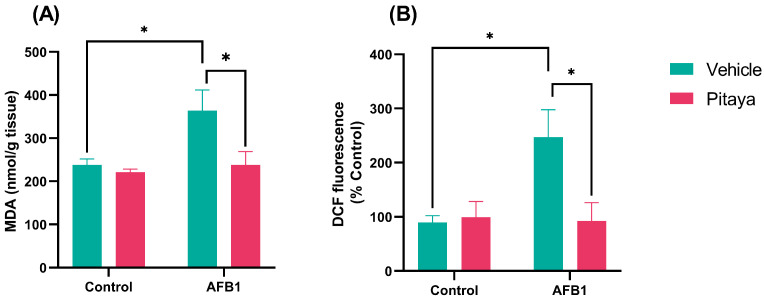
Effect of Pitaya juice pre-treatment on levels of (**A**) TBARS and (**B**) Reactive species (RS), in the liver of rats exposed to AFB1. The significance was determined through variance analysis (ANOVA, two-way) with post hoc Tukey’s range test. Data are expressed as mean ± S.E.M. (n = 4–5). The values are considered significant when *p* < 0.05. * indicates a significance level for *p* < 0.05.

**Figure 4 jof-09-00874-f004:**
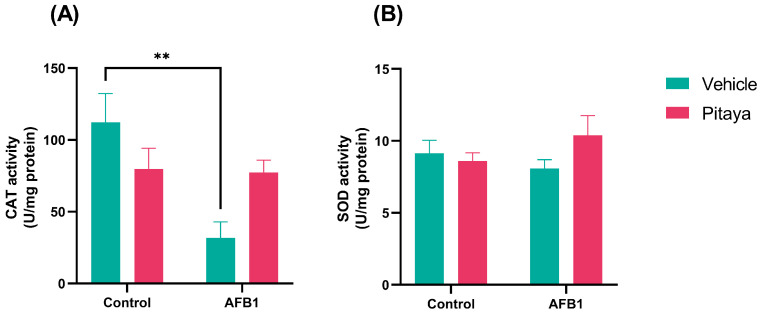
Effect of Pitaya juice pre-treatment on activity of the enzymes that defend against the attack of radicals on cells (**A**) CAT and (**B**) SOD in rats exposed to AFB1. The significance was determined through variance analysis (ANOVA, two-way) with post hoc Tukey’s range test. Data are expressed as mean ± S.E.M. (n = 4–5). The values are considered significant when *p* < 0.05. ** indicate a significance level for *p* < 0.01.

**Figure 5 jof-09-00874-f005:**
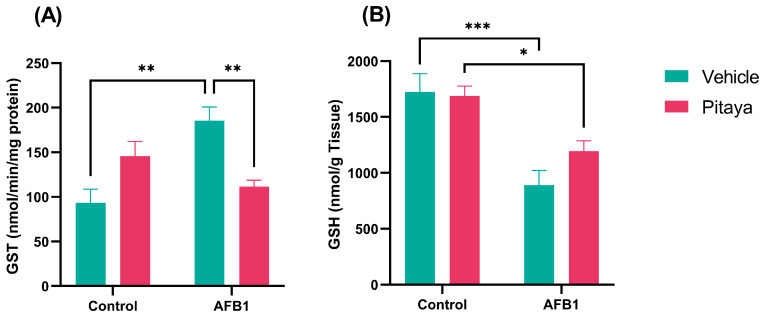
Effect of Pitaya juice pre-treatment on activity of biotransformation and elimination of xenobiotic enzymes, as well as cell defense against oxidative stress (**A**) GST and (**B**) GSH in the liver of rats exposed to AFB1. The significance was determined through variance analysis. (ANOVA, two-way) with post hoc Tukey’s range test. Data are expressed as mean ± S.E.M. (n = 4–5). The values are considered significant when *p* < 0.05. * indicates a significance level for *p* < 0.05, ** indicate a significance level for *p* < 0.01, and *** indicate a significance level for *p* < 0.001.

**Figure 6 jof-09-00874-f006:**
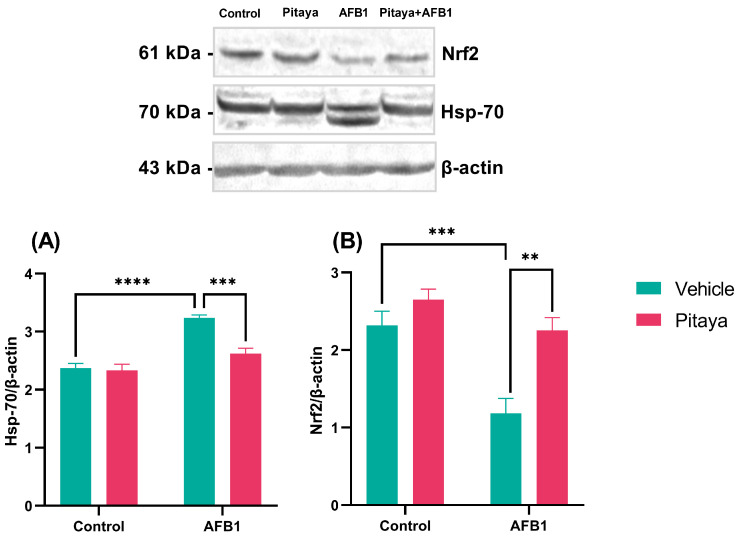
Western blot analysis of nuclear translocation of Hsp-70 (**A**) and Nrf2 (**B**) in the liver of rats exposed to AFB1. The significance was determined through variance analysis. (ANOVA, two-way) with post hoc Tukey’s range test. Data are expressed as mean ± S.E.M. (n = 4–5). The values are considered significant when *p* < 0.05. ** indicate a significance levels for *p* < 0.01, *** indicate a significance level for *p* < 0.001, and **** indicate a significant level for *p* < 0.0001.

## Data Availability

Not applicable.

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
