# Peer review of "Pitaya Juice Consumption Protects against Oxidative Damage Induced by Aflatoxin B1"

_jof, 2023, doi:10.3390/jof9090874_

Round 1
Reviewer 1 Report
Title: The paper titled "Oxidative damage aflatoxin B1 induced were protected by pitaya juice consumption" investigated the protective effect of pitaya juice against the oxidative damage and liver toxicity inflicted by exposure to AFB1 mycotoxin using rat models. Based on the experimental findings, it appears that pre-administration of pitaya juice diminishes the ALP activity in rat serum and counteracts the effects of AFB1. The juice seemed effective in mitigating alterations in TBARS, RS, Hsp-70, and Nrf2 triggered by AFB1, consequently attenuating liver injury in rats. These findings not only offer substantial evidence for the protective potential of pitaya juice against AFB1 toxicity but also lay the groundwork for future investigations to identify other potential targets of its antifungal toxicosis actions.
1. How was the dosing concentration of AFB1 determined? It would be beneficial to cite relevant literature for clarity and justification.
2. Were the serum and tissue samples centrifuged at low temperatures to ensure sample integrity?
3. There appears to be a discrepancy between the annotations in Figure 3.2 and the depicted indicators.
4. The writing style of the title might benefit from a revision for clarity. A suggested alternative: "Pitaya Juice Consumption Protects against Oxidative Damage Induced by Aflatoxin B1.
5. Please address the following typographical errors: "Hsp-70t" in line 22 seems to have an unintended "t" at the end. The abbreviation "RS" in line 72 should match its extended form, "Reactive species," mentioned elsewhere in the text. The abbreviation "AF" in line 248 should be corrected to "AFB1."
6. Considering the hepatotoxicity of AFB1, it would be pertinent to include histopathological evaluations or microscopy images to visually substantiate the liver damage and the protective effect of pitaya juice.
7. Could the authors elucidate the specific mechanism by which pitaya juice counteracts AFB1-induced oxidative damage? While the paper posits that the antioxidant compounds present in the juice—such as vitamins, flavonoids, carotenoids, and anthocyanins—play a protective role against ROS, it would be valuable to pinpoint the principal contributors. Are the absolute concentrations of flavonoids, carotenoids, and anthocyanins in pitaya juice known? A recent study published in the European Respiratory Journal has shown that flavonoids from plants and indoles from indoor environmental microorganisms play a pivotal role in countering respiratory inflammation, including conditions like asthma and wheeze (DOI: 10.1183/13993003.00260-2022). Furthermore, the study highlighted the significance of the NF-κB pathway as a potential mechanistic route. Could the protective mechanism of pitaya juice be akin to this? I recommend referencing this study and incorporating a discussion on this potential parallel in the Discussion section.
English language should be improved.
Reviewer 2 Report
Dear editor and authors,
the article titled: Oxidative damage aflatoxin B1 induced were protected by Pitaya juice consumption is relatively well written. From a quantitative point of view, quite a lot of experiments have been done (even if a simpler methodology was used), but the paper still lacks experiments that would have significantly improved it. I found many similar papers. However, it is a very interesting paper. Mycotoxicoses are really dangerous and the occurrence of aflatoxins in food is quite common, so the paper has potential. However, it seems to me that the authors spent more time discussing than proving. I would recommend a revision of the results, which seem to me to be described too simply and not all the connections are clear to me. I list below a few points to consider before further elaboration:
· - Write all p-values in italics (throughout the document).
· -I recommend including the main objectives of the thesis and a brief description of how the objectives were achieved at the end of the introduction,
· -The results of the work should be more detailed, and the explanation and description of the results should be improved.
· -In the Western blot analysis, it is not clear to me how the authors quantified the aflatoxin, please explain, also the description of figure no. 6 is more focused on the description of the statistics, which I don't see in the figure...So on what basis is your conclusion that pitaya has an effect on aflatoxin reduction?
· -Make sure the line spacing is correct throughout the paper. The graphic presentation of image 6 is poor - replace them.
· - I also recommend the authors to rewrite the conclusion, better specify the obtained results and state the limits of this study as well as future aspects.
Minor editing of English language is required. English is fine there are only minor typos in the text.
Reviewer 3 Report
The wording of the title could be improved.
In general, moderate editing of English language is required throughout manuscript.
Abstract contains many abbreviations that are not explained. Please, explain all abbreviations the first time they are used.
Line 16: Decimals in English use a period and not a comma to separate the whole number from the rest.
Lines 16 -17: Contradictory: Pitaya juice and aflatoxin B1 were administered orally (by gavage) or by injection?
Introduction
Many supporting references are too old and should be updated. Revise for typos (only a few are mentioned in this review).
Reference #4 from 2004 is too old to support that approximately 40% of diseases in emerging countries are related to aflatoxin exposure aflatoxins (which inaccurate anyway).
Line 57: typo, do you mean betalains instead of betalanas?
Line 64: do you mean AFB1?
Material and methods: I do not understand why TBARS is expressed as MDA/mg of protein, what does protein have to do with lipid peroxidation? The same for CAT, SOD and GST. Would these not be indicators of protein oxidation rather than lipid oxidation?
2.8. How many replicates were represented by the mean and SEM?
Please, in the two-way ANOVA, identify by its name the two independent variables and the dependent variable.
Results
Line 226: Please keep the same name of the experimental groups throughout the manuscript. The AFB1 + vehicle group, which one is equivalent to those defined in the M&M section (lines 99-103)? The same for others. Please stick to the group names previously defined in M&M to avoid confusion. It is deduced that ‘vehicle’ is water.
Figure 3A revise label in y-axis (/protein or /mg protein?)
Figure 3B revise label in y-axis (what is DCF?)
Moderate editing of English language is required throughout manuscript.
Round 2
Reviewer 1 Report
I have no further comment.
Reviewer 2 Report
Dear editor and authors
The article has been visibly improved. I thank the authors for explaining the ambiguities regarding quantification, but I recommend to include all information in the article (which the authors provided to me as an explanation of the results and also to describe the obtained results more) and to explain it in more detail in the future (not only in two sentences), so that there are no misunderstandings. Adjust line spacing throughout the document (e.g. Section 2.6.6.) After some adjustments, I recommend publishing the article in JoF.
Some minor English language corrections are still needed.
Reviewer 3 Report
The authors have responded satisfactorily to the issues raised and the article has been corrected in a timely and appropriate manner.
Suggestion: correct to nmol/g tissue in line 146 and y-axis in Figure 3A, instead of nmol g/ tissue
